# BIG-LITTLE NET: AN EFFICIENT MULTI-SCALE FEATURE REPRESENTATION FOR VISUAL AND SPEECH RECOGNITION

**Chun-Fu (Richard) Chen, Quanfu Fan, Neil Mallinar, Tom Sercu, Rogerio Feris**
IBM T.J. Watson Research Center, Yorktown Heights, NY 10598
{chenrich, qfan, neil.r.mallinar, tom.sercu1, rsferis}@us.ibm.com

## ABSTRACT

In this paper, we propose a novel Convolutional Neural Network (CNN) architecture for learning multi-scale feature representations with good tradeoffs between speed and accuracy. This is achieved by using a multi-branch network, which has different computational complexity at different branches with different resolutions. Through frequent merging of features from branches at distinct scales, our model obtains multi-scale features while using less computation. The proposed approach demonstrates improvement of model efficiency and performance on both object recognition and speech recognition tasks, using popular architectures including ResNet, ResNeXt and SEResNeXt. For object recognition, our approach reduces computation by $1/3$ while improving accuracy significantly over 1% point than the baselines, and the computational savings can be higher up to $1/2$ without compromising the accuracy. Our model also surpasses state-of-the-art CNN acceleration approaches by a large margin in terms of accuracy and FLOPs. On the task of speech recognition, our proposed multi-scale CNNs save 30% FLOPs with slightly better word error rates, showing good generalization across domains.

## 1    INTRODUCTION

Deep Convolutional Neural Network (CNN) models have achieved substantial performance gains in many computer vision and speech recognition tasks (He et al., 2016; 2017; Vinyals et al., 2017; Sercu & Goel, 2016). However, the accuracy obtained by these models usually grows proportionally with their complexity and computational cost. This poses an issue for deploying these models in applications that require real-time inferencing and low-memory footprint, such as self-driving vehicles, human-machine interaction on mobile devices, and robotics.

Motivated by these applications, many methods have been proposed for model compression and acceleration, including techniques such as pruning (Dong et al., 2017; Li et al., 2017; Han et al., 2015), quantization (Hubara et al., 2016; Li & Liu, 2016), and low-rank factorization (Wen et al., 2017; Ioannou et al., 2015; Zhang et al., 2016). Most of these methods have been applied to *single-scale* inputs, without considering multi-resolution processing. More recently, another line of work applies dynamic routing to allocate different workloads in the networks according to image complexity (Wu et al., 2018; Wang et al., 2018; Figurnov et al., 2017; Veit & Belongie, 2018). Multi-scale feature representations have proven successful for many vision and speech recognition tasks compared to single-scale methods (Nah et al., 2017; Chen et al., 2017a; Tóth, 2017; Farabet et al., 2013); however, the computational complexity has not been addressed much in multi-scale networks.

The computational cost of a CNN model has much to do with the input image size. A model, if running at half of the image size, can gain a remarkable computational saving of 75%. Based on this fact, we propose an efficient network architecture by combining image information at different scales through a multi-branch network. As shown in Fig. 1, our key idea is to use a high-complexity branch (*accurate but costly*) for low-scale feature representation and low-complexity branch (*efficient but less accurate*) for high-scale feature representation. The two types of features are frequently merged together to complement and enrich each other, leading to a stronger feature representation than either of them individually. We refer to the deeper branch operating at low image resolution as *Big-Branch*

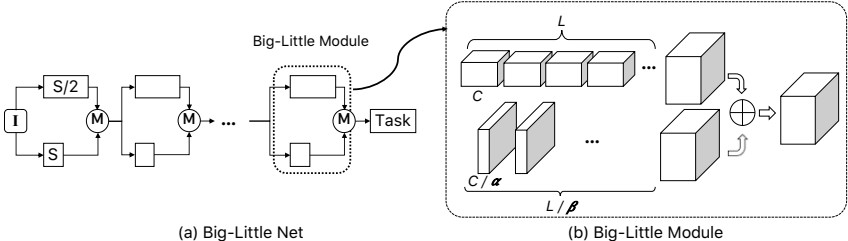

Figure 1: Our proposed Big-Little Net (*bL-Net*) for efficient multi-scale feature representations. (a) The *bL-Net* stacks several Big-Little Modules. A bL-module include $K$ branches ($K = 2$ in this illustration) where the $k^{th}$ branch represents an image scale of $1/2^k$. 'M' here denotes a merging operation. (b) Our implementation of the Big-Little Module includes two branches. The Big-Branch has the same structure as the baseline model while the Little-Branch reduces the convolutional layers and feature maps by $\alpha$ and $\beta$, respectively. Larger values of $\alpha$ and $\beta$ lead to lower computational complexity in Big-Little Net.

and the shallower one as *Little-Branch* to reflect their differences in computation. The new network architecture is thus called *Big-Little Net* or *bL-Net* for short in this paper.

While being structurally simple, our approach is quite effective. We demonstrate later that when *bL-Net* is integrated into state-of-the-art CNNs such as ResNet, ResNeXt and SEResNeXt, it yields $2\times$ computational savings over the baselines without losing any accuracy. It also outperforms many other recently developed approaches based on more sophisticated architectures at the same FLOP count. One work that relates to ours is the Inception model (Szegedy et al., 2016a; 2017), which also leverages parallel pathways in a network building block for efficiency. However, the efficiency of these models relies on substantial use of $1\times1$ and separable filters (i.e. $1\times3$ and $3\times1$). In contrast, *bL-Net* is general and applicable to many architectures including Inception.

The main contributions of our paper are summarized as follows:

- We propose an efficient and effective multi-scale CNN architecture for object and speech recognition.
- We demonstrate that our approach reduces computation by $1/3$ in models such as ResNet and ResNeXt while improving accuracy over 1% point than the baselines, and the computational savings can be higher up to $1/2$ without losing any accuracy; these results outperform state-of-the-art networks that focus on CNN acceleration by a large margin at the same FLOPs.
- We validate the proposed method on a speech recognition task, where we also achieve better word error rates while reducing the number of FLOPs by 30%.

## 2 RELATED WORK

**Model Compression and Acceleration.** Network pruning (Dong et al., 2017; Li et al., 2017) and quantization (Hubara et al., 2016; Li & Liu, 2016) are popular techniques to remove model redundancy and save computational cost. Another thread of work consists of training a sparse model directly, such as IGCv2 (Xie et al., 2018) and SCConv (Fan et al., 2017). Efficient network architectures like MobileNetV2 (Sandler et al., 2018) and ShuffleNetV2 (Ma et al., 2018) have also been explored for training compact deep networks. Other methods include knowledge distillation (Hinton et al., 2015), compression with structured matrices (Cheng et al., 2015; Sindhwani et al., 2015), and hashing (Chen et al., 2015). Dynamic routing (Wu et al., 2018; Wang et al., 2018; Figurnov et al., 2017; Veit & Belongie, 2018) has also been explored in residual networks to improve efficiency. These methods operate with single-resolution inputs, while our approach processes multi-resolution data. It could be used in tandem with these methods to further improve efficiency.

**Multi-Resolution Feature Representations.** The notion of multi-scale feature representation can be dated back to image pyramids (Adelson et al., 1984) and scale-space theory (Lindeberg & ter Haar Romeny, 1994). More recently, several methods have proposed multi-scale CNN-based architectures for object detection and recognition. MSCNN (Cai et al., 2016), DAG-CNN (Yang

& Ramanan, 2015) and FPN (Lin et al., 2017) use features at different layers to form multi-scale features. Hourglass networks (Newell et al., 2016) use a hierarchical multi-resolution model for human pose estimation. However, this approach induces a heavy workload as the complexity of each sub-network in their model is equal. Nah *et al.* (Nah et al., 2017) and Eigen *et al.* (Eigen & Fergus, 2015) combine the features from multiple networks working on different resolutions to generate multi-scale features. The overall computational cost grows along with the number of scales, leading to inefficient models. In contrast to existing methods, our approach uses different network capacities for different scales, and yields more powerful multi-scale features by fusing them at multiple levels of the network model.

Closely related to our work, approaches such as (Huang et al., 2018; Saxena & Verbeek, 2016), apply multiple branches at multi-scales while aiming at reducing computational complexity. In contrast to our work, their computational gain mostly comes from early exit depending on the input image. Our approach speeds up a network constantly regardless of the input image.

## 3  OUR APPROACH

We develop a simple, easy-to-implement, yet very efficient and effective network architecture. It learns multi-scale feature representations by fusing multiple branches with different image scales and computational complexity. As shown in Fig. 1, we design a multi-scale feature module with the following principles: (I) each branch corresponds to a single unique image scale (or resolution); (II) the computational cost of a branch is inversely proportional to the scale. Note that the principle (II) implies that we use high-complexity networks at lower resolutions and low-complexity networks at higher resolutions for the sake of efficiency.

### 3.1  BIG-LITTLE NET

Big-Little Net is a sequence of Big-Little Modules, each one taking input $\mathbf{x}_i$ and producing output $\mathbf{x}_{i+1}$. Within a Big-Little Module, assume we have $K$ branches working on $K$ scales $[1, 1/2, 1/4, \ldots, 1/2^{K-1}]$. We denote a feature map $\mathbf{x}_i$ at scale $1/2^k$ as $\mathbf{x}_i^k$, indicating the spatial size of $\mathbf{x}_i$ downsampled by $2^k$ with respect to the original input dimension. We use a weighted sum to combine all branches into a feature representation at scale $1/2^k$. Mathematically, the module's output $\mathbf{x}_{i+1}$ can be expressed by

$$\mathbf{x}_{i+1} = F\left(\sum_{k=0}^{K-1} c^k S^k\left(f_k\left(\mathbf{x}_i^k\right)\right)\right), \tag{1}$$

where $f_k(\cdot)$ denotes a sequence of convolutional layers. Typically for higher $k$, $f_k$ will have more convolutional layers having more feature maps. $S^k(\cdot)$ is the operation that matches the output size of the branches, either: (1) increasing the number feature maps with a $1 \times 1$ convolution, (2) upsampling to match the output size of the $k = 0$ branch, or both. $c^k$ indicates the weighting coefficients of each scale in the merge while $F(\cdot)$ is an optional final fusion layer like a convolutional layer.

Note that branches are merged at the end of every Big-Little Module and merging the branch outputs happens at the highest resolution and highest number of feature maps between the branches. Maintaining these large intermediate states avoids information loss. A crucial aspect of this design is that through consecutive merging and downsampling, the expensive branches operating at low resolution still have access to the high resolution information, processed by the cheaper branches in the previous module.

While our design is suitable for any number of networks, in this work we primarily focus on the case of two networks, i.e., $K = 2$. We also experimented with $K > 2$ in object and speech recognition; however, the $K = 2$ case provided the best balance between accuracy and computation (See Appendix A.4 and Section 4.2 for details). Following the principles above, we propose a multi-network architecture that integrates two branches for multi-scale feature representation. Figure 1 (b) shows an example Big-Little Net architecture.

The module includes two branches, each of which represents a separate network block from a deep model (*accurate but costly*) and a less deep counterpart (*efficient but less accurate*). The two branches are fused at the end through linear combination with unit weights (i.e., $c^0 = c^1 = 1.0$). Before fusion,

the low resolution feature maps are upsampled using bilinear interpolation to spatially match the higher-resolution counterparts ($=S^1(\cdot)$). Similarly, the high resolution feature map has an additional $1 \times 1$ convolutional layer to increase the number of output channels ($=S^0(\cdot)$). Furthermore, since our design is based on ResNet, we add a residual block to further fuse the combined features (i.e., $F(\cdot)$ is a residual block). For convenience, we refer to these two branches as *Big-Branch* (more layers and channels at low resolution) and *Little-Branch* (fewer layers and channels at high resolution), respectively. We also denote the module as *Big-Little Module* and the entire architecture as *Big-Little Net* or *bL-Net*.

To control the complexity of *bL-Net*, we introduce two parameters to specify the complexity of the Little-Branch with respect to the Big-Branch. The Big-Branch typically follows the structure of the original network, but the Little-Branch needs to be heavily slimmed and shortened to reduce computation as it operates on high resolution. Here we use two parameters $\alpha$ and $\beta$ to control the width and depth of the Little-Branch, respectively. As shown in Fig. 1 (b), $\alpha$ specifies the reduction factor of the number of channels in the convolutional layers of Little-Branch with respect to that of the original network while $\beta$ is the reduction factor of the number of convolutional layers. Larger values of $\alpha$ and $\beta$ lead to lower complexity in *bL-Net*. As demonstrated later, with an appropriate choice of $\alpha$ and $\beta$ (See Table 1 and Table 3), the cost of the Little-Branch can be $1/6$ of the *bL-Net* and $1/12$ of the original network while still providing sufficient complementary information to the Big-Branch.

## 3.2 NETWORK MERGING

We consider two options for merging the outputs of the branches. The first option is a linear combination, which joins features from two networks by addition (Newell et al., 2016). The alternative concatenates the outputs of the two networks along the channel dimension, and if needed, a $1 \times 1$ convolution can be subsequently applied to reduce the number of feature maps (Szegedy et al., 2015). Both merging approaches have their pros and cons. With linear combination, the branches can easily compensate each other, meaning each branch can activate output neurons not activated by the other. However, additional cost is added as both the size of feature maps and the number of channels in the two branches need to be adjusted to be the same before addition. On the other hand, merging by concatenation only needs to spatially align the feature maps. On the other hand, concatenation only needs to align the feature map size, however requires a $1 \times 1$ convolution reducing the number of channels after concatenation, which is a more expensive operation than the pointwise addition.

While linear combination provides an immediate exchange of the activations of both branches, concatenation relies on the following layers for this exchange. This delay in exchange could possibly be problematic if the information from each branch is destructively altered before merging. For example, a nonlinearity such as ReLU would discard all activations less than zero, effectively ignoring negative features in both branches before merging. Since linear combination does not cause too much overhead and provides better accuracy, we chose linear combination as our merging approach. In Appendix A.4, we empirically show that the linear combination approach performs better than concatenation in object recognition.

## 4 EXPERIMENTAL RESULTS

We conducted extensive experiments, as discussed below, to validate the effectiveness of our proposed *bL-Net* on object and speech recognition tasks. *bL-Net* can be easily integrated with many modern CNNs and here we chose ResNet (He et al., 2016) as the primary architecture to evaluate our approach. For simplicity, from now on, we denote by bL-*M* the *bL-Net* using a backbone network *M*. For example, bL-ResNet-50 is the Big-Little net based on ResNet-50.

## 4.1 OBJECT RECOGNITION

We used the ImageNet dataset (Russakovsky et al., 2015) for all the experiments below on object recognition. This dataset is a common benchmark for object recognition, which contains 1.28 million training images and 50k validation images with labels from 1000 categories. The details of our experimental setup and the network structures for bL-ResNet-50, 101 and 152 can be found in Appendix A.1.

Table 1: Complexity study of the Little-Branch ($\alpha$ and $\beta$) for bL-ResNet-50.

| Model | Top-1 Error | FLOPs ($10^9$) | Params ($10^6$) |
|---|---|---|---|
| ResNet-50 | 23.66% | 4.09 | 25.55 |
| bL-ResNet-50 ($\alpha = 2$, $\beta = 2$) | 22.72% | 2.91 (1.41$\times$) | 26.97 |
| bL-ResNet-50 ($\alpha = 2$, $\beta = 4$) | **22.69%** | 2.85 (1.44$\times$) | 26.69 |
| bL-ResNet-50 ($\alpha = 4$, $\beta = 2$) | 23.20% | 2.49 (1.64$\times$) | 26.31 |
| bL-ResNet-50 ($\alpha = 4$, $\beta = 4$) | 23.15% | **2.48 (1.65$\times$)** | 26.24 |

Table 2: Performance comparison for bL-ResNet, bL-ResNeXt and bL-SEResNeXt.

| Model | Top-1 Error | FLOPs ($10^9$) | Params ($10^6$) | Speed (ms/batch)[†] |
|---|---|---|---|---|
| ResNet-101 | 21.95% | 7.80 | 44.54 | 186 |
| bL-ResNet-101 ($\alpha = 2$, $\beta = 4$) | 21.80% | **3.89 (2.01$\times$)** | 41.85 | 140 (1.33$\times$) |
| bL-ResNet-101@256 ($\alpha = 2$, $\beta = 4$) | **21.04%** | 5.08 (1.54$\times$) | 41.85 | 162 (1.15$\times$) |
| ResNet-152 | 21.51% | 11.51 | 60.19 | 266 |
| bL-ResNet-152 ($\alpha = 2$, $\beta = 4$) | 21.16% | **5.04 (2.28$\times$)** | 57.36 | 178 (1.49$\times$) |
| bL-ResNet-152@256 ($\alpha = 2$, $\beta = 4$) | **20.34%** | 6.58 (1.75$\times$) | 57.36 | 205 (1.30$\times$) |
| ResNeXt-50 (32$\times$4d) | 22.20% | 4.23 | 25.03 | 157 |
| bL-ResNeXt-50 (32$\times$4d) ($\alpha = 2$, $\beta = 4$) | 21.60% | **3.03 (1.40$\times$)** | 26.19 | 125 (1.26$\times$) |
| bL-ResNeXt-50@256 ($\alpha = 2$, $\beta = 4$) | **20.96%** | 3.95 (1.08$\times$) | 26.19 | 153 (1.03$\times$) |
| ResNeXt-101 (32$\times$4d) | 21.20% | 7.97 | 44.17 | 269 |
| bL-ResNeXt-101 (32$\times$4d) ($\alpha = 2$, $\beta = 4$) | 21.08% | **4.08 (1.95$\times$)** | 41.51 | 169 (1.59$\times$) |
| bL-ResNeXt-101@256 ($\alpha = 2$, $\beta = 4$) | **20.48%** | 5.33 (1.50$\times$) | 41.51 | 203 (1.33$\times$) |
| ResNeXt-101 (64$\times$4d) | 20.73% | 15.46 | 83.46 | 485 |
| bL-ResNeXt-101 (64$\times$4d) ($\alpha = 2$, $\beta = 4$) | 20.48% | **7.14 (2.17$\times$)** | 77.36 | 263 (1.98$\times$) |
| bL-ResNeXt-101@256 (64$\times$4d) ($\alpha = 2$, $\beta = 4$) | **19.65%** | 9.32 (1.66$\times$) | 77.36 | 318 (1.53$\times$) |
| SEResNeXt-50 (32$\times$4d) | 21.78% | 4.23 | 27.56 | 216 |
| bL-SEResNeXt-50 (32$\times$4d) ($\alpha = 2$, $\beta = 4$) | 21.44% | **3.03 (1.40$\times$)** | 28.77 | 163 (1.33$\times$) |
| bL-SEResNeXt-50@256 (32$\times$4d) ($\alpha = 2$, $\beta = 4$) | **20.74%** | 3.95 (1.08$\times$) | 28.77 | 192 (1.03$\times$) |
| SEResNeXt-101 (32$\times$4d) | 21.00% | 7.97 | 48.96 | 376 |
| bL-SEResNeXt-101 (32$\times$4d) ($\alpha = 2$, $\beta = 4$) | 20.87% | **4.08 (1.95$\times$)** | 45.88 | 235 (1.60$\times$) |
| bL-SEResNeXt-101@256 (32$\times$4d) ($\alpha = 2$, $\beta = 4$) | **19.87%** | 5.33 (1.50$\times$) | 45.88 | 270 (1.39$\times$) |

[†]: speed is benchmarked on NVIDIA Tesla K80 with batch size 16. Except for @256, speed is evaluated under image size 224$\times$224. We trained all the ResNet, ResNeXt and SEResNeXt models by ourselves, so the accuracy is slightly different from the papers.

**ResNet as the backbone network.** We experimented with different complexity control factors ($\alpha$ and $\beta$) to better understand their effects on performance. $\alpha$ and $\beta$ control both the structural and computational complexity of the Little-Branch, which determines the overall computational cost of *bL-Net*.

As can be seen in Table 1, all the models based on ResNet-50 yield better performance over the baseline with less computation, clearly demonstrating the advantage of combining low- and high-complexity networks to balance between speed and accuracy. In addition, the small performance gaps between these models suggest that a computationally light Little-Branch ($< 15\%$ of the entire network) can compensate well for the low resolution representation by providing finer image details. We consider $\alpha = 2$ and $\beta = 4$ as the default setting for the following experiments. Furthermore, there are more ablation studies on the design of *bL-Net* in the Appendix A.4.

We further evaluated our approach on deeper models by using ResNet-101 and ResNet-152 as the backbone networks. We see from Table 2 that bL-ResNet-101 and bL-ResNet-152 behave similarly to bL-ResNet-50. As expected, both of them produce better results against the baseline models and achieving notable computational gains. Interestingly, our approach computationally favors deeper models, as evidenced by the fact that more speedups are observed on bL-ResNet-152 (2.3$\times$) than on bL-ResNet-101 (2.0$\times$) and bL-ResNet-50 (1.4$\times$). This is mainly because the Little Branch operating on low resolution spends less computation in a deeper model.

**ResNeXt and SEResNeXt as the backbone network.** We extended *bL-Net* to ResNeXt and SEResNeXt, two of the more accurate yet compact network architectures. We also experimented with (SE)ResNeXt-50 and (SE)ResNeXt-101 using the 32$\times$4d and 64$\times$4d setting (Xie et al., 2017; Hu et al., 2018). In our case, the Big-Branch follows the same setting of (SE)ResNeXt; however, we changed the cardinality of the Little-Branch to align with the input channels of a group convolution in the Big-Branch. All the results are shown in Table 2.

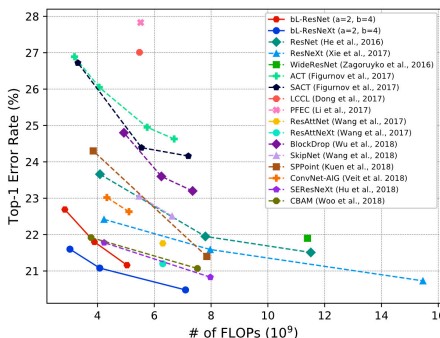

Figure 2: Comparison with the ResNet and ResNeXt related works.

bL-ResNeXt-50 achieves a moderate speedup (1.40×) and provides an additional gain of 0.6% in accuracy. However, bL-ResNeXt-101 (32×4d) gains a much more substantial speedup of 2× and seeing 0.12% improvement in accuracy. The same trend can be seen on bL-ResNeXt-101 (64×4d). On the other hand, our bL-SEResNeXts also produce better performance while reducing more FLOPs than SEResNeXts. bL-SEResNeXt-101 achieves 2× speedups and improves accuracy by 0.13%.

Since our *bL-Net* saves more budget in computation, we can evaluate a model at a larger image scale for better performance, e.g., $256 \times 256$. As illustrated in Table 2, our models evaluated at $256 \times 256$ is consistently better than their corresponding baselines while still using fewer FLOPs. The advantage becomes more pronounced with deeper models. For instance, with 40% reduction on FLOPs, bL-ResNeXt-101@256 (64×4d) boosts the top-1 performance by 1.1%, which is quite impressive given that ResNeXt-101 (64×4d) is a very competitive baseline. Table 2 also shows the running times of *bL-Net* on GPU, which indicate the practical speedups of these models are consistent with the theoretical FLOP reductions reported in Table 2.

### 4.1.1 COMPARISON WITH RELATED WORK

We first compared our method with the approaches that aim to accelerate ResNets or ResNeXts using techniques such as network pruning and adaptive computations. The results are shown in Figure 2.

Our *bL-Net* significantly outperforms all related works regarding FLOPs reduction and accuracy. Our bL-ResNet-101 is $\sim 5\%$ better than the network pruning approaches such as PFEC (Li et al., 2017) and LCCL (Dong et al., 2017), but still using less computation. When compared to SACT and ACT (Figurnov et al., 2017), our bL-ResNet-101 improves the accuracy by 5% while using the same number of FLOPs. On the other hand, our bL-ResNet-101 outperforms some of the most recent works including BlockDrop (Wu et al., 2018), SkipNet (Wang et al., 2018) and ConvNet-AIG (Veit & Belongie, 2018) by 3.7%, 2.2%, 1.2% top-1 accuracy at the same FLOPs, respectively. This clearly demonstrates the advantages of a simple fusion of two branches at different scales over the more sophisticated dynamic routing techniques developed in these approaches. In comparison to SPPoint (Kuen et al., 2018) under the same FLOPs, our bL-ResNet-101 surpasses it by 2.2% in accuracy.

We also compared *bL-Net* with the variants of ResNets, like ResAttNe(X)t (Wang et al., 2017), SEResNeXt (Hu et al., 2018) and CBAM (Woo et al., 2018). These models introduces attention mechanisms to enhance feature representations in either the channel or the spatial domain. From Figure 2, it can be seen that our *bL-Net* outperforms all of them in both FLOPs and accuracy. Our *bL-Net* achieves better performance than ResAttNeXt while saving 1.5× FLOPs. It also surpasses $\sim 1\%$ point with similar FLOPs or the similar accuracy with $\sim 1.7\times$ FLOPs reduction for both SEResNeXt and CBAM. It's worth noting that *bL-Net* can be potentially integrated with these models to further improve their accuracy and efficiency.

Finally, we ran a benchmark test on bL-ResNeXt@256 under the PyTorch framework with a batch size of 16 on a K80 GPU, and compared against various models that are publicly available. These models include Inception-V3 (Szegedy et al., 2016a), Inception-V4 (Szegedy et al., 2017), Inception-ResNet-V2 (Szegedy et al., 2017), PolyNet (Zhang et al., 2017), NASNet (Zoph et al., 2018), PNASNet (Liu

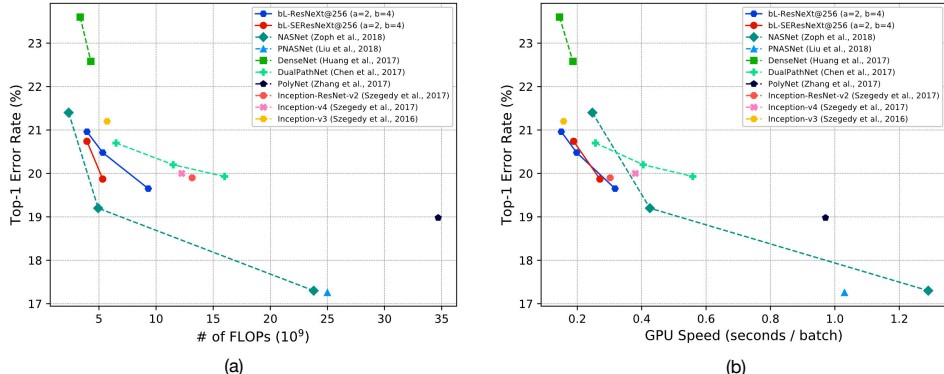

Figure 3: Comparison performance among other types of networks. (a) FLOPs. (b) GPU Speed.

et al., 2018), DualPathNet (Chen et al., 2017b) and DenseNet (Huang et al., 2017b). Among them, NASNet currently achieves the best accuracy on ImageNet.

From Fig. 3, we can see that overall, our bL-ResNeXt gains a better tradeoff between efficiency and accuracy. Compared to the Inception networks, *bL-Net* are better in both FLOPs and GPU running time. The bL-ResNeXt is also 2% point better than DenseNet at the same running speed, and $2\times$ faster than DualPathNet at the same performance.

NASNet achieves lower FLOPs and higher accuracy than *bL-Net*; however, the networks result in slow GPU running time since their operations are divided into small pieces (i.e. a fragmented structure), which are not friendly for parallel computations. On the other hand, *bL-Net*, although requiring more FLOPs, can still enjoy the computation optimization brought by modern GPU cards. Moreover, compared with the lowest FLOP configuration of NASNet, our bL-ResNeXt is 0.5% point better while running $1.5\times$ faster.

To further validate the generality of *bL-Net*, we also integrated the *bL-Net* with the highly efficient network, ShuffleNetV2 (Ma et al., 2018), and the results can be found in Appendix A.5. We also demonstrate the adaptability of *bL-Net* on the object detection task (see Appendix A.6).

## 4.2 SPEECH RECOGNITION

We train ResNet style acoustic models in the hybrid framework on Switchboard+Fisher (2000h) and provide results on Hub5 (Switchboard and Call Home portions). Switchboard is a large dataset with 2000 hours of transcribed speech from $28,000$ speakers, which is actively used as benchmark (Xiong et al., 2016; Saon et al., 2017) akin to ImageNet in the computer vision community. Our ResNet acoustic models are similar to the state of the art models described in (Saon et al., 2017), though slightly simplified (less fully connected layers) and trained with a simpler procedure (no class balancing). We provide results only after Cross-Entropy training and after decoding with a small language model (4M n-grams). Gains from this setting are typically maintained in the standard further pipelines like fine-tuning with sequence training, using more complex language models.

Appendix B gives a thorough overview of the architecture of the speech acoustic models. The main difference speech acoustic models have compared to image classification networks, is that striding or pooling only happens along the *frequency* axis, while along in the time direction we need to output dense predictions per frame (Sercu & Goel, 2016). This means that the branches at different resolutions have a fundamentally different view of the signal as it is propagating through the network; the ratio of resolution in frequency (downsampled in the Big-Branch) vs resolution in time (same between branches) is different. We can think about this as the convolutional kernels having different "aspect ratios" between branches. Therefore we not only expect FLOP reductions in *bL-Net*, but expect to have increased representational power. In addition, similar to the case in object recognition (Table 2), we could process the speech signal at higher frequency resolution than what is computationally feasible for the baseline ResNets.

Table 3 shows the results for the different architectures described in Appendix B. Most results are in line with the observations in the object recognition *bL-Net*. When comparing the baseline ResNet-22

Table 3: Speech recognition results. We present results on Hub5 and the CallHome portion of Hub5, while the RT-02 Switchboard set was used for selecting decode epoch and HMM prior settings.

| | Model | FLOPs ($10^9$) | Params ($10^6$) | WER Avg | Hub5 | Hub5 CH |
|---|---|---|---|---|---|---|
| 1 | Baseline: ResNet-22 | 1.11 | 3.02 | 14.67% | 11.15% | 18.17% |
| 2 | bL-ResNet-22 ($\alpha = 4, \beta = 1$) | 0.68 (1.63×) | 3.15 | 14.72% | 11.24% | 18.18% |
| 3 | bL-ResNet-22 ($\alpha = 4, \beta = 2$) | 0.66 (1.68×) | 3.11 | 14.47% | **10.95%** | 17.95% |
| 4 | bL-ResNet-22 ($\alpha = 4, \beta = 3$) | **0.65 (1.70×)** | 3.10 | 14.66% | 11.25% | 18.05% |
| 5 | bL-ResNet-22 ($\alpha = 2, \beta = 3$) | 0.77 (1.43×) | 3.07 | **14.46%** | 11.10% | **17.80%** |
| 6 | bL-ResNet-22 ($\alpha = 4, \beta = 1$) cat | 0.70 (1.58×) | 3.18 | 14.67% | 11.31% | 18.00% |
| 7 | bL-PYR-ResNet-22 ($\alpha = 4, \beta = 1$) | 0.98 (1.13×) | 3.32 | 14.50% | 11.05% | 17.92% |

(line 1) to the best bL-ResNet-22 (line 5), we see not only a reduction in FLOPs, but also a modest gain in Word Error Rate (WER). Comparing lines 2-4, we see that increasing $\beta$ (i.e. shorter little branches at full resolution) causes no WER degradation, while reducing the number of FLOPs. From line 5 we see that, similar to the object recognition ResNet results, decreasing $\alpha$ from 4 to 2 (i.e. keeping more feature maps in the full-resolution little branches) is important for performance, even though this increases the FLOPs again. We can summarize the best setting of $\alpha = 2$ and $\beta = 3$ for the little branches at full resolution: make them shorter but with more feature maps. This is consistent with the image classification results. From line 2 vs. line 6, the concatenation merge mode performs similar to the default additive merging, while increasing the number of FLOPs. Line 7 (compare to line 2) shows an experiment with additional branches on the lower layers (See Appendix B). Although there is some gain in WER, the added parameters and compute on the lower layers may not make this a worthwhile trade-off.

## 4.3 DISCUSSION ON *bL-Net*

From the results of both tasks, we observe the following common insights, which enable us to design an efficient multi-scale network with competitive performance: (I) The Little-Branch can be very light-weight, (II) *bL-Net* performs better when the Little-Branch is wide and shallow (smaller $\alpha$ and larger $\beta$), (III) merging is effective when the feature dimension has changed, and (IV) branch merging by addition is more effective than concatenation. (I) is because the Big-Branch can extract essential information, a light Little-Branch is good enough to provide sufficient information the Big-Branch lacks. Regarding (II), wider networks have been shown to perform better than deep networks while using a similar number of parameters. (III) is well-discussed in Appendix A.4. Finally (IV), merging through addition provides better regularization for both branches to learn complementary features to form strong features.

## 5 CONCLUSION

We proposed an efficient multi-scale feature representation based on integrating multiple networks for object and speech recognition. The Big-Branches gain significant computational reduction by working at low-resolution input but still extract meaningful features while the Little-Branch enriches the features from high-resolution input but with light computation. On object recognition task, we demonstrated that our approach provides approximately 2× speedup over baselines while improving accuracy, and the result significantly outperforms the state-of-the-art networks by a large margin in terms of accuracy and FLOPs reduction. Furthermore, when using the proposed method on speech recognition task, we gained 0.2% WER and saved 30% FLOPs at the same time. In pratice, the proposed *bL-Net* shows that the reduced FLOPs can consistently speed up the running time on GPU. That evidence showed that the proposed *bL-Net* is an efficient multi-scale feature representation structure for competitive performance with less computation. In this paper, we chose ResNet, ResNeXt and SEResNeXt as our backbone networks but *bL-Net* can be integrated with other advanced network structures, like DenseNet (Huang et al., 2017b), DualPathNet (Chen et al., 2017b) and NASNet (Zoph et al., 2018) to achieve competitive performance while saving computations. Furthermore, *bL-Net* can be integrated with those CNN acceleration approaches to make models more compact and efficient.

ACKNOWLEDGMENTS

The authors would like to thank Dr. Paul Crumley and Dr. I-Hsin Chung for their help with the hardware infrastructure setup.

Quanfu Fan and Rogerio Feris are supported by IARPA via DOI/IBC contract number D17PC00341. The U.S. Government is authorized to reproduce and distribute reprints for Governmental purposes notwithstanding any copyright annotation thereon. Disclaimer: The views and conclusions contained herein are those of the authors and should not be interpreted as necessarily representing the official policies or endorsements, either expressed or implied, of IARPA, DOI/IBC, or the U.S. Government.

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

# Appendix

The appendix illustrates the details of our experiments on object recognition and speech recognition and more ablation study.

## A  *bL-Net* FOR OBJECT RECOGNITION

### A.1  EXPERIMENTAL SETUP

We used the ImageNet dataset for all experiments. We trained all the models by Tensorpack (Wu, 2017), a higher-level wrapper for Tensorflow (Abadi et al., 2015). All the models were trained with 110 epochs, batch size 256, weight decay 0.0001, momentum 0.9 and Nesterov momentum optimizer. Furthermore, we used cosine learning-rate schedule as (Huang et al., 2017a; Loshchilov & Sgdr, 2017). We deployed the popular augmentation technique in (Szegedy et al., 2016b; Gross & Wilber, 2016) to increase the variety of training data, and randomly crop a $224 \times 224$ patch as training image. The validation error is evaluated by resizing the shorter side of an image to 256 and then crop a $224 \times 224$ from the center. Note that the results reported here for the vanilla ResNet, ResNeXt and SEResNeXt models are difference from those reported in the original paper (He et al., 2016; Gross & Wilber, 2016; Hu et al., 2018). Our vanilla ResNet is better than the original paper while vanilla ResNeXt and SEResNeXt is slightly worse than the original paper.

Table 4: Network configurations of bL-ResNet-50. Output size is illustrated in the parenthesis.

| Layers | bL-ResNet-50 | | ResNet-50 |
|---|---|---|---|
| Convolution | $7 \times 7, 64, s2$ (112 × 112) | | |
| bL-module | $3 \times 3, 64, s2$ | $\begin{pmatrix} 3\times3, 32 \\ 3\times3, 32, s2 \\ 1\times1, 64 \end{pmatrix}$ (56 × 56) | MaxPooling (56 × 56) |
| bL-module | ResBlock$_B$, 256 ×2  ResBlock$_L$, 128 ×1 
 ResBlock, 256, s2 (28 × 28) | | ResBlock, 256 ×3, s1 
 (56 × 56) |
| bL-module | ResBlock$_B$, 512 ×3  ResBlock$_L$, 256 ×1 
 ResBlock, 512, s2 (14 × 14) | | ResBlock, 512 ×4, s2 
 (28 × 28) |
| bL-module | ResBlock$_B$, 1024 ×5  ResBlock$_L$, 512 ×1 
 ResBlock, 1024 (14 × 14) | | ResBlock, 1024 ×6, s2 
 (14 × 14) |
| ResBlock | ResBlock, 2048 ×3, s2 (7 × 7) | | |
| Average pool | $7 \times 7$ average pooling | | |
| FC, softmax | 1000 | | |

ResBlock$_B$: the first $3 \times 3$ convolution is with stride 2, and a bi-linear upsampling is applied at the end.
ResBlock$_L$: a $1 \times 1$ convolution is applied at the end to align the channel size.
s2: the stride is set to 2 for the $3 \times 3$ convolution in the ResBlock.

### A.2  NETWORK STRUCTURE

This section shows the details of network structures of our bL-ResNet, and the setting of $\alpha$ and $\beta$ is 2 and 4, respectively. To understand how do we design bL-ResNet-50 based on ResNet-50, Table 4 shows the details of network structure. We used a bottleneck residual block as a ResBlock, and a $ResBlock, C$ denotes a block composed of $1 \times 1$, $3 \times 3$, and $1 \times 1$ convolutions, where the first $1 \times 1$ and the $3 \times 3$ have $C/4$ kernels and the last $1 \times 1$ has $C$ kernels.

First, the Big-Branch and the Little-Branch shares a residual block at the transition layer, so the number of residual blocks in each branch will be subtracted by 1. The number of residual blocks in the Little-Branch is defined as $\lceil \frac{L}{\beta} \rceil - 1$ and at least one, where $L$ is the number of residual blocks in the Big-Branch, and the number of kernels in a convolutional layer would be $\frac{C}{\alpha}$, where $C$ is the number of kernels in the Big-Branch. Thus, for all stages, the number of blocks in the Little-Branch

is only one, and the number of blocks in the Big-Branch would be the number of blocks in ResNet-50 miuns one.

For bL-ResNet-101 and bL-ResNet-152, we redistributed the residual blocks to different stages to balance the residual blocks at each stage. A ResNet model has 5 stages, and each stage has the same spatial size. These two models accumulate most of the convolutions (or computations) on the $4^{th}$ stage, where the size of feature maps is $14 \times 14$ when input size is $224 \times 224$. While such a design may be suitable for a very deep model, it likely limits the ability of Big-Branch to learn information at large scales, which mostly resides at earlier stages. Thus, we move some blocks in the $4^{th}$ stage of these two models to the $2^{nd}$ and $3^{rd}$ stages. Table 5 shows the details of bL-ResNet-101 and bLResNet-152.

Table 5: Network configurations of bL-ResNets, and $\alpha = 2$ and $\beta = 4$.

| Layers | Output Size | bL-ResNet-101 | | bL-ResNet-152 | |
|---|---|---|---|---|---|
| Convolution | $112 \times 112$ | $7 \times 7, 64, s2$ | | | |
| bL-module | $56 \times 56$ | $3 \times 3, 64, s2$ | | $\begin{pmatrix} 3\times3, 32 \\ 3\times3, 32, s2 \\ 1\times1, 64 \end{pmatrix}$ | |
| bL-module | $56 \times 56$ | $\begin{pmatrix} 1\times1, 64 \\ 3\times3, 64 \\ 1\times1, 256 \end{pmatrix}_B \times 3(2)$ | $\begin{pmatrix} 1\times1, 32 \\ 3\times3, 32 \\ 1\times1, 128 \end{pmatrix}_L \times 1$ | $\begin{pmatrix} 1\times1, 64 \\ 3\times3, 64 \\ 1\times1, 256 \end{pmatrix}_B \times 4(2)$ | $\begin{pmatrix} 1\times1, 32 \\ 3\times3, 32 \\ 1\times1, 128 \end{pmatrix}_L \times 1$ |
| transition layer | $28 \times 28$ | $\begin{pmatrix} 1\times1, 64 \\ 3\times3, 64, s2 \\ 1\times1, 256 \end{pmatrix} \times 1$ | | | |
| bL-module | $28 \times 28$ | $\begin{pmatrix} 1\times1, 128 \\ 3\times3, 128 \\ 1\times1, 512 \end{pmatrix}_B \times 7(3)$ | $\begin{pmatrix} 1\times1, 64 \\ 3\times3, 64 \\ 1\times1, 256 \end{pmatrix}_L \times 1$ | $\begin{pmatrix} 1\times1, 128 \\ 3\times3, 128 \\ 1\times1, 512 \end{pmatrix}_B \times 11(7)$ | $\begin{pmatrix} 1\times1, 64 \\ 3\times3, 64 \\ 1\times1, 256 \end{pmatrix}_L \times 2$ |
| transition layer | $14 \times 14$ | $\begin{pmatrix} 1\times1, 128 \\ 3\times3, 128, s2 \\ 1\times1, 512 \end{pmatrix} \times 1$ | | | |
| bL-module | $14 \times 14$ | $\begin{pmatrix} 1\times1, 256 \\ 3\times3, 256 \\ 1\times1, 1024 \end{pmatrix}_B \times 17(22)$ | $\begin{pmatrix} 1\times1, 128 \\ 3\times3, 128 \\ 1\times1, 512 \end{pmatrix}_L \times 3$ | $\begin{pmatrix} 1\times1, 256 \\ 3\times3, 256 \\ 1\times1, 1024 \end{pmatrix}_B \times 29(35)$ | $\begin{pmatrix} 1\times1, 128 \\ 3\times3, 128 \\ 1\times1, 512 \end{pmatrix}_L \times 6$ |
| transition layer | $14 \times 14$ | $\begin{pmatrix} 1\times1, 256 \\ 3\times3, 256 \\ 1\times1, 1024 \end{pmatrix} \times 1$ | | | |
| ResBlock | $7 \times 7$ | $\begin{pmatrix} 1\times1, 512 \\ 3\times3, 512, s2 \\ 1\times1, 2048 \end{pmatrix} \times 3$ | | | |
| Average pool | $1 \times 1$ | $7 \times 7$ average pooling | | | |
| FC, softmax | | 1000 | | | |

For each $B$ block, the first $3 \times 3$ convolution is with stride 2, and a bi-linear upsampling is applied at the end.
For each $L$ block, a $1 \times 1$ convolution is applied at the end.
s2: the stride is set to 2 for the convolutional layer.
The number in the parethesis denotes the original number of blocks in ResNet.

### A.3 PERFORMANCE ON LOW RESOLUTION INPUT

We analyzed what advantages *bL-Net* could provide as compared to the network which works on low resolution input directly (ResNet-50-*lowres*). As shown in Table 6, ResNet-50-*lowres* reduces lots of computations but its accuracy is not acceptable; however, bL-ResNet-50 ($\alpha = 2$ and $\beta = 4$) achieves a better balance between accuracy and performance. A similar trend is also observed on a deeper model ResNet-101-*lowres*. While such performance is unsatisfying compared to the state of the art, it is quite reasonable and expected given that almost $3 \sim 4\times$ reduction of computation are achieved in such a case.

Figure 4 shows the prediction results from bL-ResNet-50 and ResNet-50-*lowres*. When both models predict correctly (4 (a) and (b)), the bL-ResNet-50 provides better confidence for the prediction. Because the object only occupies a small portion of an image, the Little-Branch can still capture the object clearly. On the other hand, when the key features of an object is small, like the shape of beak of a bird (c) and the spots of a ladybug (d), bL-ResNet-50 can easily retain that key feature to predict correctly while ResNet-50-*lowres* provides wrong predicted label.

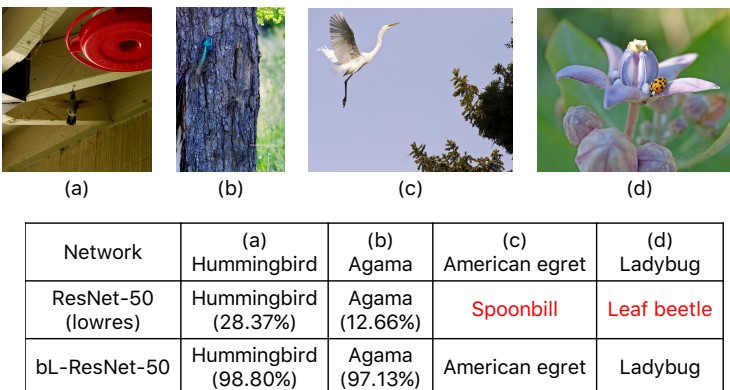

| Network | (a)
Hummingbird | (b)
Agama | (c)
American egret | (d)
Ladybug |
|---|---|---|---|---|
| ResNet-50
(lowres) | Hummingbird
(28.37%) | Agama
(12.66%) | Spoonbill | Leaf beetle |
| bL-ResNet-50 | Hummingbird
(98.80%) | Agama
(97.13%) | American egret | Ladybug |

Figure 4: Prediction results for bL-ResNet-50 and ResNet-50-*lowres*. True labels, predicted labels and their probability are listed in the table. When both models predicts correctly ((a) and (b)), bL-ResNet-50 achieves much higher probability; on the other hand, bL-ResNet-50 captures the details on the object and then predicts correctly ((c) and (d)).

Table 6: Performance of ResNets at different input resolutions.

| Network | Top-1 Error | FLOPs ($10^9$) | Params ($10^6$) |
|---|---|---|---|
| ResNet-50 | 23.66% | 4.09 | 25.55 |
| ResNet-50-*lowres* | 26.10% | 1.29 (3.17$\times$) | 25.60 |
| ResNet-101 | 21.95% | 7.80 | 44.54 |
| ResNet-101-*lowres* | 24.80% | 2.22 (3.51$\times$) | 44.57 |

### A.4 ABLATION STUDY ON NETWORK MERGING AND MULTI-BRANCH

**Is linear combination better than concatenation?** We adopt the simpler addition in *bL-Net*. Nonetheless, if we design the Big-Branch in a way that the output channels is identical to the backbone networks, then the number of kernels in the Big-Branch would be only $1-\alpha$ with respect to the total number of kernels of the backbone network; thus, in this case, the overall *bL-Net* can be more efficient while the performance degradation could be compromised. We compared the performance of these two different merging schemes in Table 7. Although concatenation approach is more efficient, it performs much worse than addition with a gap of almost $1.5\%$. This leaves addition as a better choice for *bL-Net* in both visual and speech tasks.

**More-branch in *bL-Net*** As mentioned in Section 3.1, our approach can be extended to a scenario with multiple image scales. We experimented with three scales [1/4, 1/2, 1] on bL-ResNet-50 ($K = 3$) where ResNet-50 is served as the Big-Branch at the scale of 1/4 of the original input, i.e. $56 \times 56$. As indicated in Table 7, a 3-scale *bL-Net* requires more FLOPs and parameters due to the fact that the overhead in merging more branches is significant for ResNet-50, but even though, it still cannot provide superior performance of a 2-scale *bL-Net*. This is because the Big-Branch in the 3-scale *bL-Net* is downsampled aggressively by $4$ times, thus substantially degrade the capability of feature representation in the Big-Branch.

Table 7: Different scales and merging schemes on bL-ResNet.

| Network | Top-1 Error | FLOPs ($10^9$) | Params ($10^6$) |
|---|---|---|---|
| ResNet-50 | 23.66% | 4.09 | 25.55 |
| bL-ResNet-50 (addition, $K = 2$) | **22.69%** | 2.85 (1.43$\times$) | 26.69 |
| bL-ResNet-50 (concatenation, $K = 2$) | 24.04% | 2.01 (2.03$\times$) | 20.57 |
| bL-ResNet-50 (addition, $K = 3$) | 24.12% | 3.91 (1.04$\times$) | 27.23 |

Table 8: Different number of merges in bL-ResNet. $m$: number of merges. ($\alpha = 2$, $\beta = 4$)

| Model | Top-1 Error | FLOPs ($10^9$) | Params ($10^6$) |
|---|---|---|---|
| bL-ResNet-50 ($m = 4$) (baseline) | **22.69%** | 2.85 | 26.69 |
| bL-ResNet-50 ($m = 2$) | 23.48% | 2.74 | 26.66 |
| bL-ResNet-50 ($m = 1$) | 24.57% | **2.64** | 26.64 |
| bL-ResNet-101 ($m = 4$) (baseline) | **21.80%** | **3.89** | 41.85 |
| bL-ResNet-101 ($m = 7$) | 21.85% | 5.21 | 44.44 |

**Number of Merges in *bL-Net***

We also analyzed the number of merges we needed in the *bL-Net*. One big difference between our approach and others is that *bL-Net* merges multiple times as opposed to only once in most of the other approaches. Below we provide an explanation of why more information exchange is encouraged in our approach and when is the best moment for merging operation.

In the above *bL-Net*, we merged branches before the feature dimension changes, except for the first stride convolution; thus, we used 4 merges ($m = 4$). We experimented with a different number of merges for bL-ResNet-50 and bL-ResNet-101, and the results are shown in Table 8. Since there are fewer layers in bL-ResNet-50, we reduce the number of merges to show their importance; on the other hand, there are more layers in bL-ResNet-101, so we add more merges to show that those additional merges would similarly not improve the performance anymore.

The accuracy of the bL-ResNet-50 models with less number of merges ($m = 1$ and $m = 2$) is significantly worse than with more ($m = 4$) and they do not save many FLOPs and parameters at all. This justifies frequent information exchange improves the performance. On the other hand, bL-ResNet-101 ($m = 7$) uses more merges; however, it also does not improve the performance and requires more FLOPs, which comes from more merges. This is because the original setting for the amount of merging happened when either the channel number or feature map size is changed, so extra merges happened at the feature dimension. Thus, those extra merges could be redundant since merging at identical dimension could be reduced to one merging. Hence, it empirically proves that merging before dimension is changed is the most effective.

## A.5   *bL-Net* FOR HIGH-EFFICIENCY NETWORK

To demenstrate *bL-Net* can be applied on different types of network, we deploy *bL-Net* on the high-efficiency network, ShuffleNetV2 (Ma et al., 2018), and results are shown in Table 9. Our bL-ShuffleNetV2@256 outperforms ShuffleNetV2 by up to 0.5% point under similar FLOPs, which suggests that our approach can also improve the high-efficiency networks.

Table 9: Comparison with ShuffleNetV2 (Ma et al., 2018) ($\alpha = 2$, $\beta = 2$).

| Model | Model width | Top-1 Error | FLOPs ($10^6$) | Params ($10^6$) |
|---|---|---|---|---|
| ShuffleNetV2 | 1× | 30.60% | 146 | 2.30 |
| | 1.5× | 28.03% | 299 | 3.51 |
| | 2× | 26.85% | 588 | 7.40 |
| bL-ShuffleNetV2@256 | 1× | 30.84% | 150 | 2.33 |
| | 1.5× | 27.83% | 298 | 4.80 |
| | 2× | 26.38% | 590 | 7.60 |

All models are trained by ourselves.

## A.6   *bL-Net* FOR OBJECT DETECTION

We demonstrate the effectiveness of *bL-Net* on object detection. We use *bL-Net* as a backbone network for FasterRCNN+FPN (Lin et al., 2017) on the PASCAL VOC (Everingham et al., 2010) and MS COCO datasets (Lin et al., 2014). Table 10 shows the comparison with the detector with ResNet-101

as the backbone network, and the results show that *bL-Net* achieves competitive performance while saving about $1.5\times$ FLOPs[1], suggesting that *bL-Net* is transferable to other vision tasks.

Table 10: Objection detection results.
Detection performance on the PASCAL VOC 2007test dataset.

| Network | mAP@[IoU=0.5] (bbox) | FLOPs$^{\dagger}$ $(10^9)$ |
|---|---|---|
| ResNet-101 | 81.5 | 137.70 |
| bL-ResNet-101 ($\alpha$=2, $\beta$=4) | 81.4 | 89.21 (1.54$\times$) |
| bL-ResNet-101 ($\alpha$=2, $\beta$=2) | 81.5 | 93.95 (1.47$\times$) |

Detection performance on the MS COCO val2017 dataset.

| Network | mAP@[IoU=0.50:0.95] (bbox) | FLOPs$^{\ddagger}$ $(10^9)$ |
|---|---|---|
| ResNet-101 | 39.2 | 234.85 |
| bL-ResNet-101 ($\alpha$=2, $\beta$=4) | 39.5 | 151.11 (1.55$\times$) |
| bL-ResNet-101 ($\alpha$=2, $\beta$=2) | 40.4 | 160.21 (1.47$\times$) |

$\dagger$: FLOPs is calculated when the size of input image is 600$\times$1024 with 300 proposals.

$\ddagger$: FLOPs is calculated when the size of input image is 800$\times$1344 with 300 proposals.

### A.7 COMPARISON OF MEMORY REQUIREMENT

We benchmarked the GPU memory consumption in runtime at both the training and test phases for all the models evaluated in Fig. 3. The results are shown in Fig. 5. The batch size was set to 8, which is the largest number allowed for NASNet on a P100 GPU card. The image size for any model in this benchmark experiment is the same as that used in the experiment reported in Fig. 3. For *bL-Net*, the input image size is 224$\times$224 in training and 256$\times$256 in test.

From Fig. 5, we can see that *bL-Net* is the most memory-efficient for training among all the approaches. In test, bL-ResNeXt consumes more memory than inception-resnet-v2 and inception-v4 at the same accuracy, but bL-SEResNeXt outperforms all the approaches. Note that NASNet and PNASNet are not memory friendly. This is largely because they are trained on a larger image size (331$\times$331) and these models are composed of many layers.

---

[1]The saving is slightly less than that for the classification task because more computations are required outside the feature extractor.

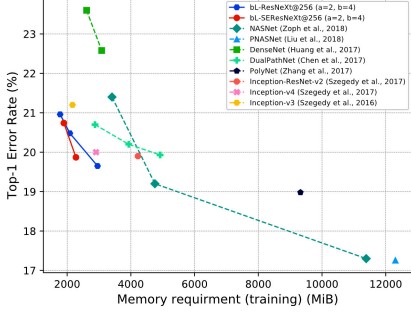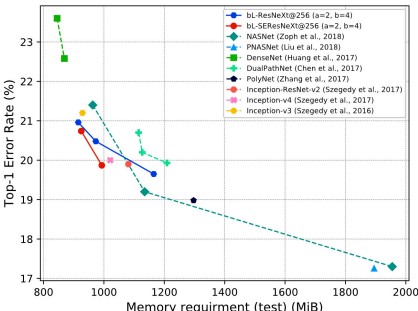

Figure 5: Comparison memory requirement at the training and test phases among other types of networks. (a) Training. (b) Test. (The batch size is 8.)

Table 11: Complexity study of the Little-Branch ($\alpha$ and $\beta$) for bL-ResNet-50 and bL-ResNet-101.

| Model | Top-1 Error | FLOPs ($10^9$) | Params ($10^6$) |
|---|---|---|---|
| ResNet-50 | 23.66% | 4.09 | 25.55 |
| bL-ResNet-50 ($\alpha = 1$, $\beta = 1$) | **21.75%** | 5.65 | 34.12 |
| bL-ResNet-50 ($\alpha = 1$, $\beta = 2$) | 22.11% | 4.34 | 30.14 |
| bL-ResNet-50 ($\alpha = 2$, $\beta = 2$) | 22.72% | 2.91 | 26.97 |
| bL-ResNet-50 ($\alpha = 2$, $\beta = 4$) | 22.69% | 2.85 | 26.69 |
| bL-ResNet-50 ($\alpha = 4$, $\beta = 2$) | 23.20% | 2.49 | 26.31 |
| bL-ResNet-50 ($\alpha = 4$, $\beta = 4$) | 23.15% | **2.48** | **26.24** |
| ResNet-101 | 21.95% | 7.80 | 44.54 |
| bL-ResNet-101 ($\alpha = 1$, $\beta = 1$) | **20.31%** | 10.29 | 63.32 |
| bL-ResNet-101 ($\alpha = 2$, $\beta = 2$) | 21.40% | 4.27 | 43.39 |
| bL-ResNet-101 ($\alpha = 2$, $\beta = 4$) | 21.80% | **3.89** | **41.85** |

# B  *bL-Net* FOR SPEECH RECOGNITION

## B.1  EXPERIMENTAL SETUP

In our experiments, we start with an input size of $64 \times 49$, where $64$ is the number of logmel filterbanks, calculated for each utterance on-the-fly. We also stack their first and second derivatives to get 3 input channels, resulting in our final input of dimensionality `batch_size` $\times 3 \times 64 \times 49$. Our output is of size `batch_size` $\times 512 \times 4 \times 1$, which is then projected to `batch_size` $\times 512 \times 1 \times 1$ and finally to `batch_size` $\times 32\text{k} \times 1 \times 1$ for classification. We then perform softmax cross-entropy over this output space of 32k tied CD states from forced alignment, doing phone prediction on the central frame of the input utterance. We report results after Cross-Entropy training, on Hub5'00 (SWB and CH part) after decoding using the standard small 4M n-gram language model with a 30.5k word vocabulary.

All models were trained in PyTorch (Paszke et al., 2017) over 16 epochs on 2 GPUs with per-GPU batch size 256 (total batch size of 512), gradient clipping 10.0, weight decay $1 \times 10^{-6}$, nesterov accelerated momentum 0.9, and learning rate 0.03 (annealed by $\sqrt{0.5}$ per epoch $\geq 10$).

## B.2  NETWORK STRUCTURES

**ResNet-22** Our models follow a ResNet architecture without padding in time (Saon et al., 2017), which accounts for the fact that padding in time adds undesirable artifacts when processing a longer utterance (Sercu & Goel, 2016). Under this constraint, each convolution operation reduces our input sequence in time by $k - 1$, where $k$ is the kernel width used. This effect can be seen in Table 12, in which the time variable, $T$, is reduced in accordance with the number of convolutions. For similar reasons, when we stride we only do so in frequency, and not in time. For the rest of this section, when we refer to striding we are referring only to striding in frequency. We define our residual blocks as a series of $3 \times 3$ convolutions. When we transition from one stage to the next, we stride by 2 on the first convolution of the following block. All of our models start with a $5 \times 5$ convolution with stride 2, which downsamples the input from 64 melbins to 32.

Our baseline model, ResNet-22, consists of four stages of two-convolution residual blocks: $(3 \times 3, 64) \times 3$; $(3 \times 3, 128, s2) \times 3$; $(3 \times 3, 256, s2) \times 3$; $(3 \times 3, 512, s2) \times 2$. The output then goes through a bottleneck projection layer to the 32k-dimensional output CD state: $(4 \times 1, 512)$ and $(1 \times 1, 32\text{k})$.

**bL-ResNet-22 ($\alpha = 4, \beta = 1$)** Table 12 displays two *bL-Net* architectures that we experimented with based on the ResNet-22 baseline. The *bL-Net* baseline, bL-ResNet-22 ($\alpha = 4, \beta = 1$), consists of two branches in each *Big-Little Module* and is well-defined through the parameters $\alpha$ and $\beta$. In between each *Big-Little Module* we downsample our input using a transition layer consisting of a residual block with a single $3 \times 3$ convolution with stride 2. Therefore, we shift one convolution operation out of the last residual block in each stage that precedes a transition layer.

Whenever downsampling is performed in the Big-Branch, it is only in the frequency dimension and not in time. Similarly, bilinear upsampling only occurs in the frequency dimension. All *bL-Net* variants end with the same projection and output layer as ResNet-22. All merges, unless otherwise specified, are through linear combination with unit weights per branch. For comparison, we experimented with a version of bL-ResNet-22 ($\alpha = 4, \beta = 1$) using concatenation to merge branches, the results of

Table 12: Network configurations of bL-ResNets applied to speech for acoustic modeling.

| Layers | Output Size | bL-ResNet-22 ($\alpha = 4, \beta = 1$) | bL-ResNet-22 ($\alpha = 2, \beta = 3$) |
|---|---|---|---|
| Convolution | $32 \times T$ $T = 49 \to 45$ | $5 \times 5, 64, s2$ | |
| bL-module | $32 \times T$ $T = 45 \to 35$ | $\begin{pmatrix} 3\times3, 64 \\ 3\times3, 64 \end{pmatrix}_B \times 2 \quad \begin{pmatrix} 3\times3, 16 \\ 3\times3, 16 \end{pmatrix}_L \times 2$ 

 $(3\times3, 64)_B \times 1 \quad (3\times3, 16)_L \times 1$ | $\begin{pmatrix} 3\times3, 64 \\ 3\times3, 64 \end{pmatrix}_B \times 2 \quad \begin{pmatrix} 3\times3, 32 \\ 3\times3, 32 \end{pmatrix}_L \times 1$ 

 $(3\times3, 64)_B \times 1$ |
| transition layer | $16 \times T$ $T = 35 \to 33$ | $(3\times3, 64, s2) \times 1$ | |
| bL-module | $16 \times T$ $T = 33 \to 23$ | $\begin{pmatrix} 3\times3, 128 \\ 3\times3, 128 \end{pmatrix}_B \times 2 \quad \begin{pmatrix} 3\times3, 32 \\ 3\times3, 32 \end{pmatrix}_L \times 2$ 

 $(3\times3, 128)_B \times 1 \quad (3\times3, 32)_L \times 1$ | $\begin{pmatrix} 3\times3, 128 \\ 3\times3, 128 \end{pmatrix}_B \times 2 \quad \begin{pmatrix} 3\times3, 64 \\ 3\times3, 64 \end{pmatrix}_L \times 1$ 

 $(3\times3, 128)_B \times 1$ |
| transition layer | $8 \times T$ $T = 23 \to 21$ | $(3\times3, 128, s2) \times 1$ | |
| bL-module | $8 \times T$ $T = 21 \to 11$ | $\begin{pmatrix} 3\times3, 256 \\ 3\times3, 256 \end{pmatrix}_B \times 2 \quad \begin{pmatrix} 3\times3, 64 \\ 3\times3, 64 \end{pmatrix}_L \times 2$ 

 $(3\times3, 256)_B \times 1 \quad (3\times3, 64)_L \times 1$ | $\begin{pmatrix} 3\times3, 256 \\ 3\times3, 256 \end{pmatrix}_B \times 2 \quad \begin{pmatrix} 3\times3, 128 \\ 3\times3, 128 \end{pmatrix}_L \times 1$ 

 $(3\times3, 256)_B \times 1$ |
| transition layer | $4 \times T$ $T = 11 \to 9$ | $(3\times3, 256, s2) \times 1$ | |
| bL-module | $4 \times T$ $T = 9 \to 1$ | $\begin{pmatrix} 3\times3, 512 \\ 3\times3, 512 \end{pmatrix}_B \times 2 \quad \begin{pmatrix} 3\times3, 128 \\ 3\times3, 128 \end{pmatrix}_L \times 2$ | $\begin{pmatrix} 3\times3, 512 \\ 3\times3, 512 \end{pmatrix} \times 2$ |
| Convolution | $1 \times 1$ | $4 \times 1, 512$ | |
| Convolution | $1 \times 1$ | $1 \times 1, 32k$ | |

For each $B$ block, the first $3 \times 3$ convolution is with stride 2 (in frequency), and a bilinear upsampling is applied at the end.

For each $L$ block, a $1 \times 1$ convolution is applied at the end to match feature maps.

$s2$: the stride is set to 2 in the frequency dimension (*not* in time) for the convolutional layer.

$T = T_0 \to T_1$ indicates that $T_0$ is the size of the time dimension at the start of the given layer, which is reduced to $T_1$ by the end of the layer.

which are presented in Table 3. Using concatenation instead of linear combination in this model results in each stage having more channels after concatenation than the current stage of the network calls for (i.e. in stage 1 we end up with $64 + 16 = 80$ channels at the end of the relevant *Big-Little Module*, whereas we only want $64$ channels to be outputted). To resolve this, we apply a $1 \times 1$ convolution to reduce the number of channels accordingly and fuse the two separate feature maps.

**bL-ResNet-22 ($\alpha = 4, \beta = 2, 3$)** We explored two more models where we fix $\alpha = 4$, one in which we take $\beta = 2$ and another where $\beta = 3$. All Big-Branchs in these models are the same as bL-ResNet-22 ($\alpha = 4, \beta = 1$). The difference in each Little-Branch is based on the setting of $\beta$. Since the number of convolutions we use in the *bL-Net* baseline is uneven in the first three stages, we take $\lceil L/\beta \rceil$ to be the depth of the Little-Branch, where $L$ is the depth of the Big-Branch. For $\beta = 2$, the first three stages of the network have a Little-Branch consisting of one residual block with two $3 \times 3$ convolutions and one residual block with one $3 \times 3$ convolution. This results in a reduction of the number of convolutions from 5 in the Big-Branch to 3 in the Little-Branch. For $\beta = 3$, the first three stages of the network have a Little-Branch consisting of one residual block with two $3 \times 3$ convolutions, resulting in a reduction in the number of convolutions from 5 in the Big-Branch to 2 in the Little-Branch. For both models, the Little-Branch in the final stage consists of a single residual block with two $3 \times 3$ convolutions, since there are only four convolutions in the Big-Branch of the final stage and $\lceil 4/3 \rceil = \lceil 4/2 \rceil = 2$.

In all *bL-Net* variants in which $\beta > 1$, because we can't pad in time, we see that the time dimension will get out of sync between the Big-Branch and Little-Branch. Therefore, before merging we need to match the output size of each branch. To do this, we crop the shallower branches in time to match the deepest branch (i.e. the Big-Branch will always have a smaller time dimension due to having more convolutions, so we crop to match it). This is similar to the way the shortcut in ResNet is dealt with in (Saon et al., 2017), and does not introduce edge artifacts when processing longer sequences.

**bL-ResNet-22 ($\alpha = 2, \beta = 3$)** The last of our two-branch models is where $\alpha = 2$ and $\beta = 3$, which is also presented in Table 12. This variant is well-defined in $\alpha$ and $\beta$ up to the first three stages

of the network. In the last stage, however, we opt to not branch and instead follow identically the final stage of ResNet-22 with two residual blocks operating at the full input resolution with $512$ channels.

**bL-PYR-ResNet-22 ($\alpha = 4, \beta = 1$)** We additionally present results on a pyramidal structure, in which the first stage of the network operates with four branches, the second with three, the third with two, and the fourth equivalent to the fourth stage of ResNet-22 (and bL-ResNet-22 ($\alpha = 2, \beta = 3$)). Due to the setting of $\alpha = 4$, we increased the number of channels in the first stage Big-Branch to have $256$ channels (with the three Little-Branches in this stage having $64, 16,$ and $4$ channels), avoiding a single channel on the smallest branch. Note that the middle branches require both resolution upsampling and $1 \times 1$ convolution to match channels. The third stage *Big-Little Module* operates on two branches and is identical to the analagous stage in bL-ResNet-22 ($\alpha = 4, \beta = 1$).

