# OpenReview forum: "Big-Little Net: An Efficient Multi-Scale Feature Representation for Visual and Speech Recognition"
_ICLR.cc/2019/Conference_

### Official Review · AnonReviewer2 · 2018-10-20
**extension of multi-scale network, and expected good results**

**Rating:** 7
**Confidence:** 4

**Review:**

The big-little module is an extension of the multi-scale module. Different scales takes different complexities: higher complexity for low-scale, and lower complexity for high scale. Two schemes of merging two branches are also discussed, and the linear combination is empirically better.

As expected, the results are better than ResNets, ResNexts, SEResNexts. I do not have  comments except ablation study is needed to show the results for more choices of alpha, beta, e.g., alpha =1, beta =1.

---

> ### Author Response · Authors · 2018-11-14
> **feedback**
>
> We thank the reviewer for the positive comments on our approach. We have included in Table 11 (Page 18) the results of bL-ResNet-50 and bL-ResNet-101 with alpha and beta both set to be 1. Not surprisingly, both models achieve the best accuracy, but they also become most costly in computation and are parameter heavy.

---

### Official Review · AnonReviewer1 · 2018-10-31
**paper review**

**Rating:** 6
**Confidence:** 5

**Review:**

The authors propose a new CNN architecture and show results on object and speech recognition. In particular, they propose a multi-scale CNN module that processes feature maps at various scales. They show compelling results on IN and a reduction of compute complexity

Pros:
(+) The paper is well written
(+) The method is elegant and reproducible
(+) Results are compelling and experimentation is thorough
Cons:
(-) Transfer to other visual tasks, beyond IN, is missing
(-) Memory requirements are not mentioned, besides FLOPs, speed and parameters

Overall, the proposed approach is elegant and clear. The impact of the multi-scale module is evident, in terms of FLOPs and performance. While their approach performs a little worse than NASNet, both in terms of FLOP efficiency and top1-error, it is simpler and easier to train. I'd like for the authors to also discuss memory requirements for training and testing the network.

Finally, various papers have appeared over the recent years showing improvements over baselines on ImageNet. However, most of these papers are not impactful, because they do not show any impact to other visual tasks, such as detection. On the contrary, methods that do transfer get adopted very fast. I would be much more convinced of this approach, if the authors showed similar performance gains (both in terms of complexity and metrics) for COCO detection.

---

> ### Author Response · Authors · 2018-11-14
> **feedback**
>
> We thank the reviewer for the constructive comments.
>
> - Transfer capability of bLNet:
> We used bLNet as a backbone network for feature extraction in the Faster RCNN + FPN detector.
> The detection results on PASCAL VOC and COCO datasets are included in Table 10 in Appendix A6.
> Our bLNet achieves comparable or better accuracy than the baseline detectors while reducing FLOPs by about 1.5 times.
> Please refer to Table 10 in Appendix A6 for more detail.
>
> - Memory requirements of bLNet:
> We benchmarked the GPU memory consumption in runtime at both the training and test phases for all the models evaluated in Fig. 3.
> The results are shown in Fig. 5 in Appendix A7. The batch size was set to 8, which is the largest number allowed for NASNet on a P100 GPU card (16 GiB memory). The image size for any model in this benchmark experiment is the same as that used in the experiment reported in Fig. 3. For bLNet, the input image size is 224x224 in training and 256x256 in test.
>
> From Fig. 5, we can see that bLNet is the most memory-efficient for training among all the approaches.
> In test, bL-ResNeXt consumes more memory than inception-resnet-v2 and inception-v4 at the same accuracy,
> but bL-SEResNeXt outperforms all the approaches. Note that NASNet and PNASNet are not memory friendly.
> This is largely because they are trained on a larger image size (331x331) and these models are composed of many layers.

---

### Official Review · AnonReviewer3 · 2018-11-03
**Simple way to gain performance and computation**

**Rating:** 7
**Confidence:** 4

**Review:**

This paper presents a novel multi-scale architecture that achieves a better trade-off speed/accuracy than most of the previous models. The main idea is to decompose a convolution block into multiple resolutions and trade computation for resolution, i.e. low computation for high resolution representations and higher computation for low resolution representations. In this way the low resolution can focus on having more layers and channels, but coarsely, while the high resolution can keep all the image details, but with a smaller representation. The branches (normally two) are merged at the end of each block with linear combination at high resolution. Results for image classification on ImageNet with different network architectures and for speech recognition on Switchboard show the accuracy and speed of the proposed model.

Pros:
- The idea makes sense and it seems GPU friendly in the sense that the FLOPs reduction can be easily converted in a real speed-up
- Results show that the joint use of two resolution can provide better accuracy and lower computational cost, which is normally quite difficult to obtain
- The paper is well written and experiments are well presented.
- The appendix shows many interesting additional experiments

Cons:
- The improvement in performance and speed is not exceptional, but steady on all models.
- Alpha and beta seem to be two hyper-parameters that need to be tuned for each layer.

Overall evaluation:
Globally the paper seems well presented, with an interesting idea and many thorough experiments that show the validity of the approach. In my opinion this paper deserves to be published.


Additional comments:
- - In the introduction (top of pag. 2) and in the contributions, the advantages of this approach are explained in a different manner that can be confusing. More precisely in the introduction the authors say that bL-Net yeald 2x computational saving with better accuracy. In the contributions they say that the savings in computation can be up to 1/2 with no loss in accuracy.

---

> ### Author Response · Authors · 2018-11-14
> **feedback**
>
> We thank the reviewer for the positive comments on our approach. We have revised the manuscript to clarify our contributions in the introduction. For the parameters alpha and beta in bLNet, although they could be tuned for each layer, we fixed them (alpha=2 and beta=4) in all our experiments except in the ablation study. We found that this universal setting in general leads to good tradeoffs between accuracy and computation cost among all the models consistently. In the future, we are interested in exploring reinforcement learning to search for optimal alpha and beta to achieve a better tradeoff.

---

### Author Response · Authors · 2018-11-14
**updated the pdf**

We updated the pdf to address the comments from the reviewers. (the revised parts are highlighted in blue.)

---

### Meta-Review · Area_Chair1 · 2018-12-17
**Simple and effective**

**Confidence:** 4
**Recommendation:** Accept (Poster)

**Metareview:**

This paper propose a novel CNN architecture for learning multi-scale feature representations with good tradeoffs between speed and accuracy. reviewers generally arrived at a consensus on accept.